# POLICY OPTIMIZATION BY GENETIC DISTILLATION

**Tanmay Gangwani**
Computer Science
UIUC
Urbana, IL 61801
gangwan2@illinios.edu

**Jian Peng**
Computer Science
UIUC
Urbana, IL 61801
jianpeng@illinois.edu

## ABSTRACT

Genetic algorithms have been widely used in many practical optimization problems. Inspired by natural selection, operators, including mutation, crossover and selection, provide effective heuristics for search and black-box optimization. However, they have not been shown useful for deep reinforcement learning, possibly due to the catastrophic consequence of parameter crossovers of neural networks. Here, we present Genetic Policy Optimization (GPO), a new genetic algorithm for sample-efficient deep policy optimization. GPO uses imitation learning for policy crossover in the state space and applies policy gradient methods for mutation. Our experiments on MuJoCo tasks show that GPO as a genetic algorithm is able to provide superior performance over the state-of-the-art policy gradient methods and achieves comparable or higher sample efficiency.

## 1 INTRODUCTION

Reinforcement learning (RL) has recently demonstrated significant progress and achieves state-of-the-art performance in games (Mnih et al., 2015; Silver et al., 2016), locomotion control (Lillicrap et al., 2015), visual-navigation (Zhu et al., 2017), and robotics (Levine et al., 2016). Among these successes, deep neural networks (DNNs) are widely used as powerful functional approximators to enable signal perception, feature extraction and complex decision making. For example, in continuous control tasks, the policy that determines which action to take is often parameterized by a deep neural network that takes the current state observation or sensor measurements as input. In order to optimize such policies, various policy gradient methods (Mnih et al., 2016; Schulman et al., 2015; 2017; Heess et al., 2017) have been proposed to estimate gradients approximately from rollout trajectories. The core idea of these policy gradient methods is to take advantage of the temporal structure in the rollout trajectories to construct a Monte Carlo estimator of the gradient of the expected return.

In addition to the popular policy gradient methods, other alternative solutions, such as those for black-box optimization or stochastic optimization, have been recently studied for policy optimization. Evolution strategies (ES) is a class of stochastic optimization techniques that can search the policy space without relying on the backpropagation of gradients. At each iteration, ES samples a candidate population of parameter vectors ("genotypes") from a probability distribution over the parameter space, evaluates the objective function ("fitness") on these candidates, and constructs a new probability distribution over the parameter space using the candidates with the high fitness. This process is repeated iteratively until the objective is maximized. Covariance matrix adaptation evolution strategy (CMA-ES; Hansen & Ostermeier (2001)) and recent work from Salimans et al. (2017) are examples of this procedure. These ES algorithms have also shown promising results on continuous control tasks and Atari games, but their sample efficiency is often not comparable to the advanced policy gradient methods, because ES is black-box and thus does not fully exploit the policy network architectures or the temporal structure of the RL problems.

Very similar to ES, genetic algorithms (GAs) are a heuristic search technique for search and optimization. Inspired by the process of natural selection, GAs evolve an initial population of genotypes by repeated application of three genetic operators - mutation, crossover and selection. One of the main differences between GA and ES is the use of the crossover operator in GA, which is able to provide higher diversity of good candidates in the population. However, the crossover operator

is often performed on the parameter representations of two parents, thus making it unsuitable for nonlinear neural networks. The straightforward crossover of two neural networks by exchanging their parameters can often destroy the hierarchical relationship of the networks and thus cause a catastrophic drop in performance. NeuroEvolution of Augmenting Topologies (NEAT; Stanley & Miikkulainen (2002b;a)), which evolves neural networks through evolutionary algorithms such as GA, provides a solution to exchange and augment neurons but has found limited success when used as a method of policy search in deep RL for high-dimensional tasks. A major challenge to making GAs work for policy optimization is to design a good crossover operator which efficiently combines two parent policies represented by neural networks and generates an offspring that takes advantage of both parents. In addition, a good mutation operator is needed as random perturbations are often inefficient for high-dimensional policies.

In this paper, we present Genetic Policy Optimization (GPO), a new genetic algorithm for sample-efficient deep policy optimization. There are two major technical advances in GPO. First, instead of using parameter crossover, GPO applies imitation learning for policy crossovers in the state space. The state-space crossover effectively combines two parent policies into an offspring or child policy that tries to mimic its best parent in generating similar state visitation distributions. Second, GPO applies advanced policy gradient methods for mutation. By randomly rolling out trajectories and performing gradient descent updates, this mutation operator is more efficient than random parameter perturbations and also maintains sufficient genetic diversity. Our experiments on several continuous control tasks show that GPO as a genetic algorithm is able to provide superior performance over the state-of-the-art policy gradient methods and achieves comparable or higher sample efficiency.

## 2 BACKGROUND AND RELATED WORK

### 2.1 REINFORCEMENT LEARNING

In the standard RL setting, an agent interacts with an environment $\mathcal{E}$ modeled as a Markov Decision Process (MDP). At each discrete time step $t$, the agent observes a state $s_t$ and choose an action $a_t \in \mathcal{A}$ using a policy $\pi(a_t|s_t)$, which is a mapping from states to a distribution over possible actions. Here we consider high-dimensional, continuous state and action spaces. After performing the action $a_t$, the agent collects a scalar reward $r(s_t, a_t) \in \mathcal{R}$ at each time step. The goal in reinforcement learning is to learn a policy which maximizes the expected sum of (discounted) rewards starting from the initial state. Formally, the objective is

$$J(\pi) = \mathbb{E}_{\{(s_t,a_t)\} \sim \mathcal{E}, \pi} \left[ \sum_{t=0}^{\infty} \gamma^t r(s_t, a_t) \right]$$

where the states $s_t$ are sampled from the environment $\mathcal{E}$ using an unknown system dynamics model $p(s_{t+1}|s_t, a_t)$ and an initial state distribution $p(s_0)$, the actions $a_t$ are sampled from the policy $\pi(a_t|s_t)$ and $\gamma \in (0, 1]$ is the discount factor.

### 2.2 POLICY GRADIENT METHODS

Policy-based RL methods search for an optimum policy directly in the policy space. One popular approach is to parameterize the policy $\pi(a_t|s_t; \theta)$ with $\theta$, express the objective $J(\pi(a_t|s_t; \theta))$ as a function of $\theta$ and perform gradient descent methods to optimize it. The REINFORCE algorithm (Williams, 1992) calculates an unbiased estimation of the gradient $\nabla_\theta J(\theta)$ using the likelihood ratio trick. Specifically, REINFORCE updates the policy parameters in the direction of the following approximation to policy gradient

$$\nabla_\theta J(\theta) \approx \sum_{t=0}^{\infty} \nabla_\theta \log \pi(a_t|s_t; \theta) R_t$$

based on a single rollout trajectory, where $R_t = \sum_{i=0}^{\infty} \gamma^i r(s_{t+i}, a_{t+i})$ is the discounted sum of rewards from time step $t$. The advantage actor-critic (A2C) algorithm (Sutton & Barto; Mnih et al., 2016) uses the state value function (or critic) to reduce the variance in the above gradient estimation. The contribution to the gradient at time step $t$ is $\nabla_\theta \log \pi(a_t|s_t; \theta)(R_t - V^\pi(s_t))$. $R_t - V^\pi(s_t)$ is an estimate of the advantage function $A^\pi(s_t, a_t) = Q^\pi(s_t, a_t) - V^\pi(s_t)$. In practice, multiple rollouts are performed to get the policy gradient, and $V^\pi(s_t)$ is learned using a function approximator.

High variance in policy gradient estimates can sometimes lead to large, destructive updates to the policy parameters. Trust-region methods such as TRPO (Schulman et al., 2015) avoid this by restricting the amount by which an update is allowed to change the policy. TRPO is a second order algorithm that solves an approximation to a constrained optimization problem using conjugate gradient. Proximal policy optimization (PPO) algorithm (Schulman et al., 2017) is an approximation to TRPO that relies only on first order gradients. The PPO objective penalizes the Kullback-Leibler (KL) divergence change between the policy before the update ($\pi_{\theta_{old}}$) and the policy at the current step ($\pi_\theta$). The penalty weight $\beta$ is adaptive and adjusted based on observed change in KL divergence after multiple policy update steps have been performed using the same batch of data.

$$J^{PPO}(\theta) = \hat{\mathbb{E}}_t \left[ \frac{\pi_\theta(a_t|s_t)}{\pi_{\theta_{old}}(a_t|s_t)} \hat{A}_t - \beta KL \left[ \pi_{\theta_{old}}(.|s_t), \pi_\theta(.|s_t) \right] \right]$$

where $\hat{\mathbb{E}}_t[...]$ indicates the empirical average over a finite batch of samples, and $\hat{A}_t$ is the advantage estimation. Schulman et al. (2017) propose another objective based on clipping of the likelihood ratio, but we use the adaptive-KL objective due to its better empirical performance (Heess et al., 2017; Hafner et al., 2017).

## 2.3 EVOLUTIONARY ALGORITHMS

There is growing interest in using evolutionary algorithms as a policy search procedure in RL. We provide a brief summary; a detailed survey is provided by Whiteson (2012). Neuroevolution is the process of using evolutionary algorithms to generate neural network weights and topology. Among early applications of neuroevolution algorithms to control tasks are SANE (Moriarty & Mikkulainen, 1996) and ESP (Gomez & Miikkulainen, 1999). NEAT has also been successfully applied for policy optimization (Stanley & Miikkulainen, 2002a). NEAT provides a rich representation for genotypes, and tracks the historical origin of every gene to allow for principled crossover between networks of disparate topologies. In Gomez et al. (2008), the authors introduce an algorithm based on cooperative co-evolution (CoSyNE) and compare it favorably against Q-learning and policy-gradient based RL algorithms. They do crossover between fixed topology networks using a multi-point strategy at the granularity of network layers (weight segments). HyperNEAT (D'Ambrosio & Stanley, 2007), which extends on NEAT and uses CPPN-based indirect encoding, has been used to learn to play Atari games from raw game pixels (Hausknecht et al., 2014).

Recently, Salimans et al. (2017) proposed a version of Evolution Strategies (ES) for black-box policy optimization. At each iteration $k$, the algorithm samples candidate parameter vectors (policies) using a fixed covariance Gaussian $\mathcal{N}(0, \sigma^2 I)$ perturbation on the mean vector $m^{(k)}$. The mean vector is then updated in the direction of the weighted average of the perturbations, where weight is proportional to the fitness of the candidate. CMA-ES has been used to learn neural network policies for reinforcement learning (CMA-NeuroES, Heidrich-Meisner & Igel (2009); Igel (2003)). CMA-ES samples candidate parameter vectors using a Gaussian $\mathcal{N}(0, C^{(k)})$ perturbation on the mean vector $m^{(k)}$. The covariance matrix and the mean vector for the next iteration are then calculated using the candidates with high fitness. Cross-Entropy methods use similar ideas and have been found to work reasonably well in simple environments (Szita & Lőrincz, 2006).

In this work, we consider policy networks of fixed topology. Existing neuroevolution algorithms perform crossover between parents by copying segments—single weight or layer(s)—of DNN parameters from either of the parents. Also, mutation is generally done by random perturbations of the weights, although more rigorous approaches have been proposed (Hansen & Ostermeier, 2001; Sehnke et al., 2010; Lehman et al., 2017). In this work, we use policy gradient algorithms for efficient mutation of high-dimensional policies, and also depart from prior work in implementing the crossover operator.

## 3 GENETIC POLICY OPTIMIZATION

### 3.1 OVERALL ALGORITHM

Our procedure for policy optimization proceeds by evolving the policies (genotypes) through a series of selection, crossover and mutation operators (Algorithm 1). We start with an ensemble of

---

**Algorithm 1** Genetic Policy Optimization

1: *population* $\leftarrow \pi_1, \ldots, \pi_m$  ▷ Initial policies with random parameters
2: **repeat**
3:    *population* $\leftarrow$ MUTATE(*population*)
4:    *parents_set* $\leftarrow$ SELECT(*population*, FITNESS-FN)
5:    *children* $\leftarrow$ empty set
6:    **for** tuple$(\pi_x, \pi_y) \in$ *parents_set* **do**
7:       $\pi_c \leftarrow$ CROSSOVER$(\pi_x, \pi_y)$
8:       add $\pi_c$ to *children*
9:    **end for**
10:    *population* $\leftarrow$ *children*
11: **until** $k$ steps of genetic optimization

---

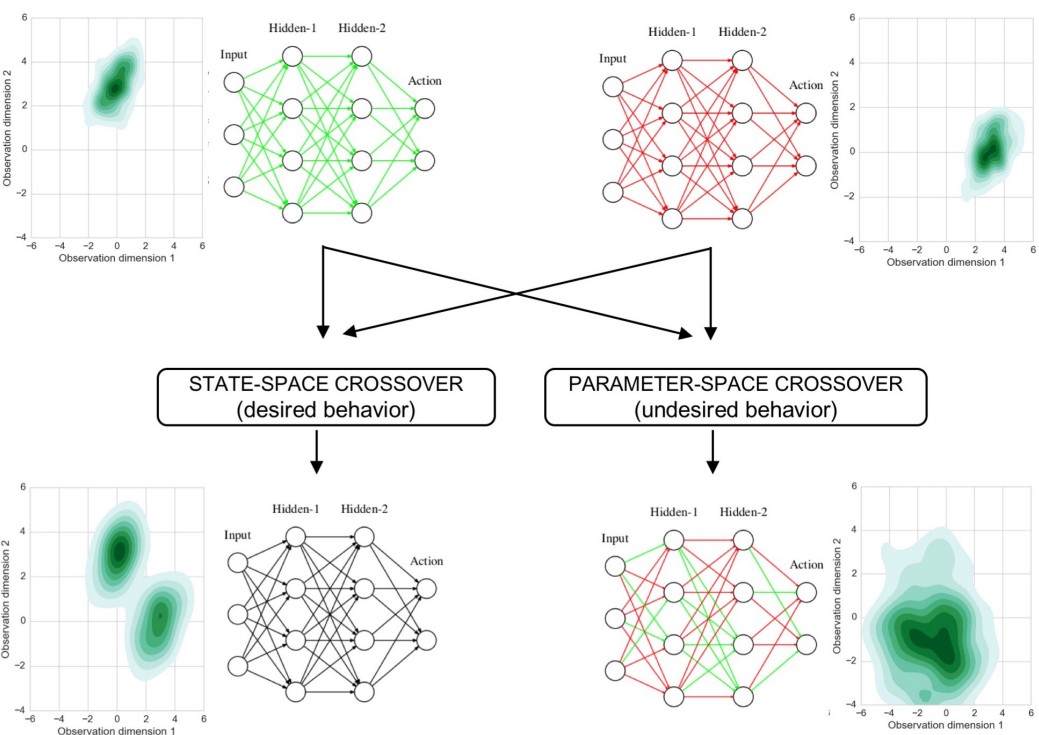

Figure 1: Different crossover strategies for neural network policies. State-visitation distribution plot next to each policy depicts the slice of state-space where that policy gives high returns. In a naïve approach like parameter-space crossover (shown in bottom-right), edge weights are copied from the parent network to create the offspring. Our proposed state-space crossover operator, instead, aims to achieve the behavior shown in bottom-left.

policies initialized with random parameters. In line 3, we mutate each of the policies separately by performing a few iterations of updates on the policy parameters. Any standard policy gradient method, such as PPO or A2C, can be used for mutation. In line 4, we create a set of parents using a selection procedure guided by a fitness function. Each element of this set is a policy-pair $(\pi_x, \pi_y)$ that is used in the reproduction (crossover) step to produce a new child policy $(\pi_c)$. This is done in line 7 by mixing the policies of the parents. In line 10, we obtain the population for the next generation by collecting all the newly created children. The algorithm terminates after $k$ rounds of optimization.

## 3.2 GPO CROSSOVER AND MUTATION

We consider policies that are parameterized using deep neural networks of fixed architectures. If the policy is Gaussian, as is common for many robotics and locomotion tasks (Duan et al., 2016), then

the network outputs the mean and the standard-deviation of each action in the action-space. Combining two DNN policies such that the final child policy possibly absorbs the *best traits* of both the parents is non-trivial. Figure 1 illustrates different crossover strategies. The figure includes neural network policies along with the state-visitation distribution plots (in a 2D space) corresponding to some high return rollouts using that policy. The two parent networks are shown in the top half of the figure. The state-visitation distributions are made non-overlapping to indicate that the parents policies have good state-to-action mapping for disparate local regions of the state-space.

A naïve approach is to do crossover in the parameter space (bottom-right in figure). In this approach, a DNN child policy is created by copying over certain edge weights from either of the parents. The crossover could be at the granularity of multiple DNN layers, a single layer of edges or even a single edge (e.g. NEAT(Stanley & Miikkulainen, 2002b)). However, this type of crossover is expected to yield a low-performance composition due to the complex non-linear interactions between policy-parameters and the expected policy return. For the same reason, the state-visitation distribution of the child doesn't hold any semblance to that of either of the parents. The bottom-left part of the figure shows the outcome of an ideal crossover in state-space. The state-visitation distribution of the child includes regions from both the parents, leading to better performance (in expectation) than either of them. In this work, we propose a new crossover operator that utilizes imitation learning to combine the best traits from both parents and generate a high-performance child or offspring policy. So this crossover is not done directly in the parameter space but in the behavior or the state visitation space. We quantify the effect of these two types of crossovers in Section 4 by mixing several DNN pairs and measuring the policy performance in a simulated environment. Imitation learning can broadly be categorized into behavioral cloning, where the agent is trained to match the action of the expert in a given state using supervised learning, and inverse reinforcement learning, where the agent infers a cost function for the environment using expert demonstrations and then learns an optimal policy for that cost function. We use behavioral cloning in this paper, and all our references to imitation learning should be taken to mean that.

Our second contribution is in utilizing policy gradient algorithms for mutation of neural network weights in lieu of the Gaussian perturbations used in prior work on evolutionary algorithms for policy search. Because of the randomness in rollout samples, the policy-gradient mutation operator also maintains sufficient genetic diversity in the population. This helps our overall genetic algorithm achieve similar or higher sample efficiency compared to the state-of-the-art policy gradient methods.

### 3.3 GENETIC OPERATORS

This section details the three genetic operators. We use different subscripts for different policies. The corresponding parameters of the neural network are sub-scripted with the same letter (e.g. $\theta_x$ for $\pi_x$). We also use $\pi_x$ and $\pi_{\theta_x}$ interchangeably. $\bigcup_{i=1}^{m} \pi_i$ represents an ensemble of $m$ policies $\{\pi_1, \ldots, \pi_m\}$.

#### 3.3.1 CROSSOVER$(\pi_x, \pi_y)$

This operator mixes two input policies $\pi_x$ and $\pi_y$ in state-space and produces a new child policy $\pi_c$. The three policies have identical network architecture. The child policy is learned using a two-step procedure. A schematic of the methodology is shown in Figure 2. Firstly, we train a two-level policy $\pi_H(a|s) = \pi_S(\text{parent} = x|s)\pi_x(a|s) + \pi_S(\text{parent} = y|s)\pi_y(a|s)$ which, given an observation, first chooses between $\pi_x$ and $\pi_y$, and then outputs the action distribution of the chosen parent. $\pi_S$ is a binary policy which is trained using trajectories from the parents involved in the crossover. In our implementation, we reuse the trajectories from the parents' previous mutation phase rather than generating new samples. The training objective for $\pi_S$ is weighted maximum likelihood, where normalized trajectory returns are used as weights. For-

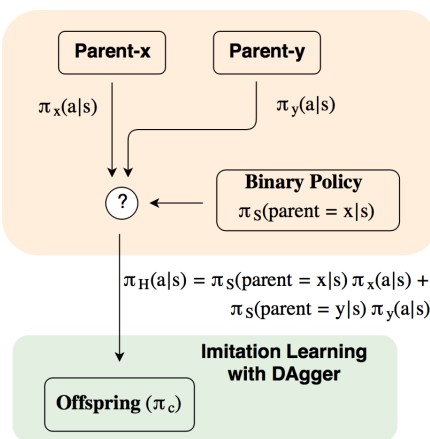

Figure 2: Schema for combining parent policies to produce an offspring policy.

mally, given two parents $\pi_x$ and $\pi_y$, the log loss is given by:

$$-\mathbb{E}_{s\sim\mathcal{D}}\Big[w_s\log\big[p_x\mathbb{I}_{s\in\tau_x}+(1-p_x)\mathbb{I}_{s\in\tau_y}\big]\Big] \tag{1}$$

where $p_x := \pi_S(\text{parent} = x|s)$, $w_s$ is the weight assigned to the trajectory which contained the state $s$, and $\mathcal{D}$ is the set of parent trajectories. This hierarchical reinforcement learning step acts a medium of knowledge transfer from the parents to the child. We use only high-reward trajectories from $\pi_x$ and $\pi_y$ as data samples for training $\pi_S$ to avoid transfer of negative behavior. It is possible to further refine $\pi_S$ by running a few iterations of any policy-gradient algorithm, but we find that the maximum likelihood approach works well in practice and can also avoid extra rollout samples.

Next, to distill the information from $\pi_H$ into a policy with the same architecture as the parents, we use imitation learning to train a child policy $\pi_c$. We use trajectories from $\pi_H$ (expert) as supervised data and train $\pi_c$ to predict the expert action under the state distribution induced by the expert. The surrogate loss for imitation learning is:

$$L^{IMIT}(\theta_c) = \mathbb{E}_{s\sim d^*}\Big[KL\big[\pi_c(.|s),\pi_H(.|s)\big]\Big] \tag{2}$$

where $d^*$ is the state-visitation distribution induced by $\pi_H$. To avoid compounding errors due to state distribution mismatch between the expert and the student, we adopt the Dataset Aggregation (DAgger) algorithm (Ross et al., 2011). Our training dataset $\mathcal{D}$ is initialized with trajectories from the expert. After iteration $i$ of training, we sample some trajectories from the current student ($\pi_c^{(i)}$), label the actions in these trajectories using the expert and form a dataset $\mathcal{D}_i$. Training for iteration $i+1$ then uses $\{\mathcal{D}\cup\mathcal{D}_1\ldots\cup\mathcal{D}_i\}$ to minimize the KL-divergence loss. This helps to achieve a policy that performs well under its own induced state distribution. The direction of KL-divergence in Equation 2 encourages high entropy in $\pi_c$, and empirically, we found this to be marginally better than the reverse direction. For policies with Gaussian actions, the KL has a closed form and therefore the surrogate loss is easily optimized using a first order method. In experiments, we found that this crossover operator is very efficient in terms of sample complexity, as it requires only a small size of rollout samples. More implementation details can be found in Appendix 6.3.

### 3.3.2 MUTATE($\bigcup\limits_{i=1}^{m}\pi_i$)

This operator modifies (in parallel) each policy of the input policy ensemble by running some iterations of a policy gradient algorithm. The policies have different initial parameters and are updated with high-variance gradients estimated using rollout trajectories. This leads to sufficient genetic diversity and good exploration of the state-space, especially in the initial rounds of GPO. For two popular policy gradient algorithms—PPO and A2C—the gradients for policy $\pi_i$ are calculated as

$$\nabla_{\theta_i}L^{PPO}(\theta_i) = \hat{\mathbb{E}}_{i,t}\left[\frac{\nabla_{\theta_i}\pi_{\theta_i}(a_t|s_t)}{\pi_{\theta_i^{(old)}}(a_t|s_t)}\hat{A}_t - \nabla_{\theta_i}\beta KL\big[\pi_{\theta_i^{(old)}}(.|s_t),\pi_{\theta_i}(.|s_t)\big]\right] \tag{3}$$

$$\nabla_{\theta_i}L^{A2C}(\theta_i) = \hat{\mathbb{E}}_{i,t}\left[\nabla_{\theta_i}\log\pi_{\theta_i}(a_t|s_t)\hat{A}_t\right] \tag{4}$$

where $\hat{\mathbb{E}}_{i,t}[...]$ indicates the empirical average over a finite batch of samples from $\pi_i$, and $\hat{A}_t$ is the advantage. We use an MLP to model the critic baseline $V^\pi(s_t)$ for advantage estimation. PPO does multiple updates on the policy $\pi_{\theta_i}$ using the same batch of data collected using $\pi_{\theta_i^{(old)}}$, whereas A2C does only a single update.

During mutation, a policy $\pi_i$ can also use data samples from other *similar* policies in the ensemble for off-policy learning. A larger data-batch (generally) leads to a better estimate of the gradient and stabilizes learning in policy gradient methods. When using data-sharing, the gradients for $\pi_i$ are

$$\nabla_{\theta_i}L^{PPO}(\theta_i) = \left(\sum_{j\in\mathbb{S}_i}\hat{\mathbb{E}}_{j,t}\left[\frac{\nabla_{\theta_i}\pi_{\theta_i}(a_t|s_t)}{\pi_{\theta_j^{(old)}}(a_t|s_t)}\hat{A}_t\right]\right) - \nabla_{\theta_i}\hat{\mathbb{E}}_{i,t}\left[\beta KL\big[\pi_{\theta_i^{(old)}}(.|s_t),\pi_{\theta_i}(.|s_t)\big]\right] \tag{5}$$

$$\nabla_{\theta_i} L^{A2C}(\theta_i) = \sum_{j \in \mathbb{S}_i} \hat{\mathbb{E}}_{j,t} \left[ \frac{\nabla_{\theta_i} \pi_{\theta_i}(a_t|s_t)}{\pi_{\theta_j^{(old)}}(a_t|s_t)} \hat{A}_t \right] \tag{6}$$

where $\mathbb{S}_i \equiv \{j \mid KL[\pi_i, \pi_j] < \epsilon$ before the start of current round of mutation$\}$ contains similar policies to $\pi_i$ (including $\pi_i$).

### 3.3.3 SELECT($\bigcup_{i=1}^{m} \pi_i$, FITNESS-FN)

Given a set of $m$ policies and a fitness function, this operator returns a set of policy-couples for use in the crossover step. From all possible $\binom{m}{2}$ couples, the ones with maximum fitness are selected. The fitness function $f(\pi_x, \pi_y)$ can be defined according two criteria, as below.

- Performance fitness as sum of expected returns of both policies, i.e. $f(\pi_x, \pi_y) \stackrel{\text{def}}{=} E_{\tau \sim \pi_x}[R(\tau)] + E_{\tau \sim \pi_y}[R(\tau)]$

- Diversity fitness as KL-divergence between policies, i.e. $f(\pi_x, \pi_y) \stackrel{\text{def}}{=} KL[\pi_x, \pi_y]$

While the first variant favors couples with high cumulative performance, the second variant explicitly encourages crossover between diverse (high KL divergence) parents. A linear combination provides a trade-off of these two measures of fitness that can vary during the genetic optimization process. In the early rounds, a relatively higher weight could be provided to KL-driven fitness to encourage exploration of the state-space. The weight could be annealed with rounds of Algorithm 1 for encouraging high-performance policies. For our experiments, we use a simple variant where we put all the weight on the performance fitness for all rounds, and rely on the randomness in the starting seed for different policies in the ensemble for diversity in the initial rounds.

## 4 EXPERIMENTS

In this section, we conduct experiments to measure the efficacy and robustness of the proposed GPO algorithm on a set of continuous control benchmarks. We begin by describing the experimental setup and our policy representation. We then analyze the effect of our crossover operator. This is followed by learning curves for the simulated environments, comparison with baselines and ablations. We conclude with a discussion on the quality of policies learned by GPO and scalability issues.

### 4.1 SETUP

All our experiments are done using the OpenAI rllab framework (Duan et al., 2016). We benchmark 9 continuous-control locomotion tasks based on the MuJoCo physics simulator [1]. All our control policies are Gaussian, with the mean parameterized by a neural network of two hidden layers (64 hidden units each), and linear units for the final output layer. The diagonal co-variance matrix is learnt as a parameter, independent of the input observation, similar to (Schulman et al., 2015; 2017). The binary policy ($\pi_S$) used for crossover has two hidden layers (32 hidden units each), followed by a softmax. The value-function baseline used for advantage estimation also has two hidden layers (32 hidden units each). All neural networks use `tanh` as the non-linearity at the hidden units. We show results with PPO and A2C as policy gradient algorithms for mutation. PPO performs 10 steps of full-batch gradient descent on the policy parameters using the same collected batch of simulation data, while A2C does a single descent step. Other hyperparameters are in Appendix 6.3.

### 4.2 CROSSOVER PERFORMANCE

To measure the efficacy of our crossover operator, we run GPO on the HalfCheetah environment, and plot the performance of all the policies involved in 8 different crossovers that occur in the first

---

[1]HalfCheetah, Walker2d, Hopper, InvertedDoublePendulum, Swimmer, Ant, HalfCheetah-Hilly, Walker2d-Hilly, Hopper-Hilly. The "Hilly" variants are more difficult versions of the original environments (`https://github.com/rll/rllab/pull/121`). We set difficulty to 1.0

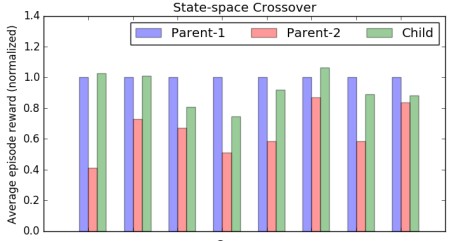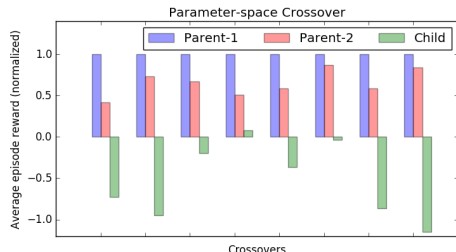

(a) Average episode reward for the child policies after state-space crossover (left) and parameter-space crossover (right), compared to the performance of the parents. All bars are normalized to the first parent in each crossover. Policies are trained on the HalfCheetah environment.

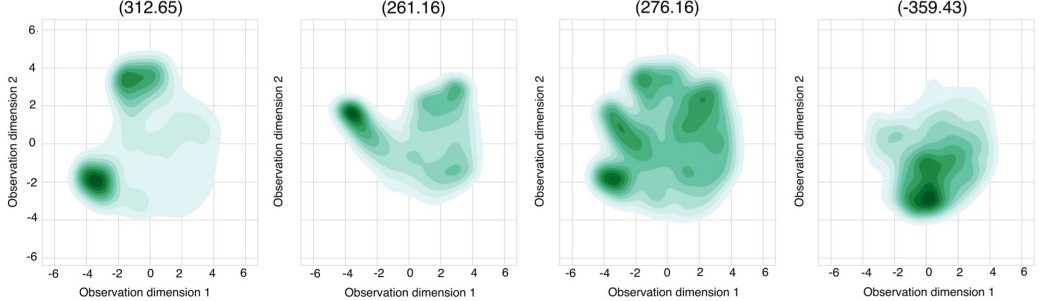

(b) State visitation distribution for high reward rollouts from policies trained on the HalfCheetah environment. From left to right - first parent, second parent, child policy using state-space crossover, child policy using parameter-space crossover. The number above each subplot is the average episode reward for 100 rollouts from the corresponding policy.

Figure 3: Comparison of Crossover operators.

| | GPO | Single | Joint |
|---|---|---|---|
| Walker2d | **1464.6 ± 93.42** | 540.93 ± 13.54 | 809.8 ± 156.53 |
| HalfCheetah | **2100.54 ± 151.58** | 1523.52 ± 45.02 | 1766.11 ± 104.37 |
| HalfCheetah-hilly | **1234.99 ± 38.72** | 661.49 ± 86.58 | 1033.44 ± 99.15 |
| Hopper-hilly | 893.69 ± 13.81 | 508.62 ± 16.47 | **904.87 ± 21.17** |
| InvertedDoublePendulum | 4647.95 ± 39.69 | **4705.72 ± 13.65** | 4539.98 ± 37.49 |
| Ant | **1337.75 ± 120.98** | 393.74 ± 18.94 | 1215.16 ± 31.18 |
| Walker2d-hilly | **1140.36 ± 146.69** | 467.37 ± 24.9 | 1044.77 ± 98.34 |
| Swimmer | **99.33 ± 0.14** | 96.29 ± 0.14 | 94.55 ± 3.6 |
| Hopper | 903.16 ± 99.37 | 457.1 ± 16.01 | **922.5 ± 61.01** |

Table 1: Mean and standard-error for final performance of GPO and baselines using PPO.

round of Algorithm 1. Figure 3a shows the average episode reward for the parent policies and their corresponding child. All bars are normalized to the first parent in each crossover. The left subplot depicts state-space crossover. We observe that in many cases, the child either maintains or improves on the better parent. This is in contrast to the right subplot where parameter-space crossover breaks the information structure contained in either of the parents to create a child with very low performance. To visualize the state-space crossover better, in Figure 3b we plot the state-visitation distribution for high reward rollouts from all policies involved in one of the crossovers. All states are projected from a 20 dimensional space (for HalfCheetah) into a 2D space by t-SNE (Maaten & Hinton, 2008). Notwithstanding artifacts due to dimensionality reduction, we observe that high reward rollouts from the child policy obtained with state-space crossover visit regions frequented by both the parents, unlike the parameter-space crossover (rightmost subplot) where the policy mostly meanders in regions for which neither of the parents have strong supervision.

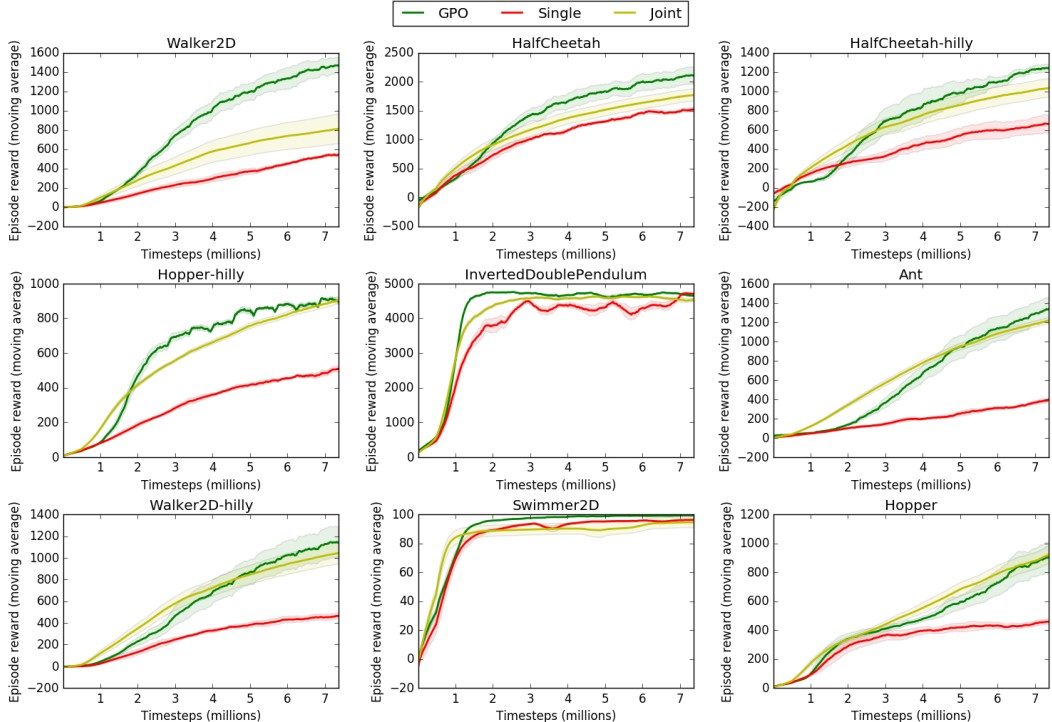

Figure 4: Performance of GPO and baselines on MuJoCo environments using PPO.

## 4.3 Comparison with Policy Gradient Methods

In this subsection, we compare the performance of policies trained using GPO with those trained with standard policy gradient algorithms. GPO is run for 12 rounds (Algorithm 1) with a population size of 8, and simulates 8 million timesteps in total for each environment (1 million steps per candidate policy). We compare with two baselines which use the same amount of data. The first baseline algorithm, `Single`, trains 8 independent policies with policy gradient using 1 million timesteps each, and selects the policy with the maximum performance at the end of training. Unlike GPO, these policies do not participate in state-space crossover or interact in any way. The second baseline algorithm, `Joint`, trains a single policy with policy gradient using 8 million timesteps. Both `Joint` and `Single` do the same number of gradient update steps on the policy parameters, but each gradient step in `Joint` uses 8 times the batch-size. For all methods, we replicate 8 runs with different seeds and show the mean and standard error.

Figure 4 plots the moving average of per-episode reward when training with PPO as the policy gradient method for all algorithms. The x-axis is the total timesteps of simulation, including the data required for DAgger imitation learning in the crossover step. We observe that GPO achieves better performance than `Single` is almost all environments. `Joint` is a more challenging baseline since each gradient step uses a larger batch-size, possibly leading to well-informed, low-variance gradient estimates. Nonetheless, GPO reaches a much better score for environments such as Walker2D and HalfCheetah, and also their more difficult (hilly) versions. We believe this is due to better exploration and exploitation by the nature of the genetic algorithm. The performance at the end of training is shown in Table 1. Results with A2C as the policy gradient method are in Appendix 6.1. With A2C, GPO beats the baselines in all but one environments. In summary, these results indicate that, with the new crossover and mutation operators, genetic algorithms could be an alternative policy optimization approach that competes with the state-of-the-arts policy gradient methods.

## 4.4 Ablations

Our policy optimization procedure uses crossover, select and mutate operators on an ensemble of policies over multiple rounds. In this section, we perform ablation studies to measure the impact

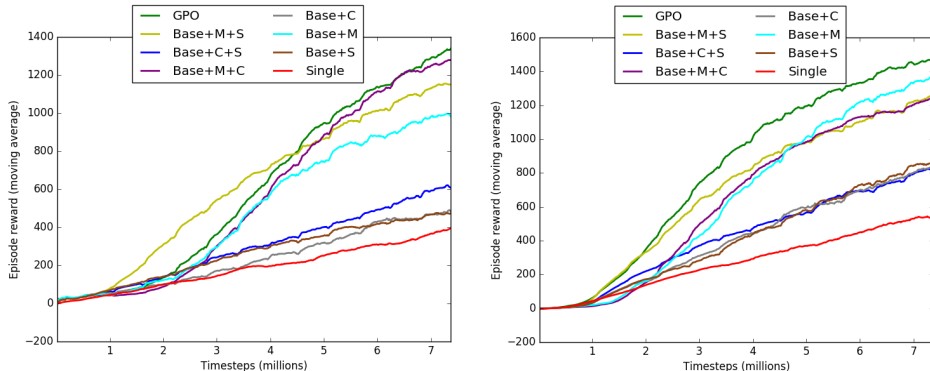

Figure 5: Ablation studies on two environments — Ant (left) and Walker-2D (right). Averaged over the two environments, the performance normalized to GPO in increasing order is Single (0.33), Base+C (0.47), Base+S (0.47), Base+C+S (0.5), Base+M (0.83), Base+M+S (0.86), Base+M+C (0.9), and GPO (1.0).

on performance when only certain operator(s) are applied. Figure 5 shows the results and uses the following symbols:

- Crossover **(C)** - Presence of the symbol indicates state-space crossover using imitation learning; otherwise a simple crossover is done by copying the parameters of the stronger parent to the offspring policy.

- Select **(S)** - Presence of the symbol denotes use of a fitness function (herein performance-fitness, Section 3.3.3) to select policy-pairs to breed; otherwise random selection is used.

- Data-sharing during Mutate **(M)** - Mutation in GPO is done using policy-gradient, and policies in the ensemble share batch samples with other similar policies (Section 3.3.2). We use this symbol when sharing is enabled; otherwise omit it.

In Figure 5, we refer to setting where none of the components {C, S, M} are used as `Base` and apply components over it. We also show `Single` which trains an ensemble of policies that do not interact in any manner. We note that each of the components applied in isolation provide some improvement over `Single`, with data-sharing (M) having the highest benefit. Also, combining two components generally leads to better performance than using either of the constituents alone. Finally, using all components results in GPO and it gives the best performance. The normalized numbers are mentioned in the figure caption.

### 4.5 ROBUSTNESS AND SCALABILITY

The SELECTION operator selects high-performing individuals for crossover in every round of Algorithm 1. Natural selection weeds out poorly-performing policies during the optimization process. In Figure 6, we measure the average episode reward for each of the policies in the ensemble at the final round of GPO. We compare this with the final performance of the 8 policies trained using the `Single` baseline. We conclude that the GPO policies are more robust. In Figure 7, we experiment with varying the population size for GPO. All the policies in this experiment use the same batch-size for the gradient steps and do the same number of gradient steps. Performance improves by increasing the population size suggesting that GPO is a scalable optimization procedure. Moreover, the MUTATE and CROSSOVER genetic operators lend themselves perfectly to multiprocessor parallelism.

## 5 CONCLUSION

We presented Genetic Policy Optimization (GPO), a new approach to deep policy optimization which combines ideas from evolutionary algorithms and reinforcement learning. First, GPO does efficient policy crossover in state space using imitation learning. Our experiments show the benefits

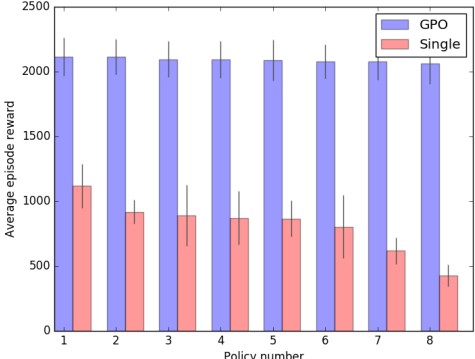

Figure 6: Final performance of the policy ensemble trained with GPO and `Single` on the HalfCheetah environment.

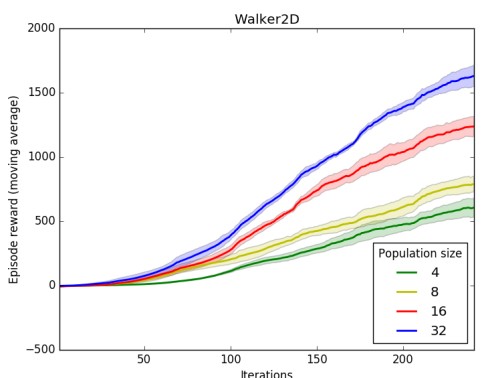

Figure 7: Scaling by increasing the GPO population size in Walker2d environment.

of crossover in state-space over parameter-space for deep neural network policies. Second, GPO mutates the policy weights by using advanced policy gradient algorithms instead of random perturbations. We conjecture that the noisy gradient estimates used by policy gradient methods offer sufficient genetic diversity, while providing a strong learning signal. Our experiments on several MuJoCo locomotion tasks show that GPO has superior performance over the state-of-the-art policy gradient methods and achieves comparable or higher sample efficiency. Future advances in policy gradient methods and imitation learning will also likely improve the performance of GPO for challenging RL tasks.

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

# 6 APPENDIX

## 6.1 PERFORMANCE WITH A2C

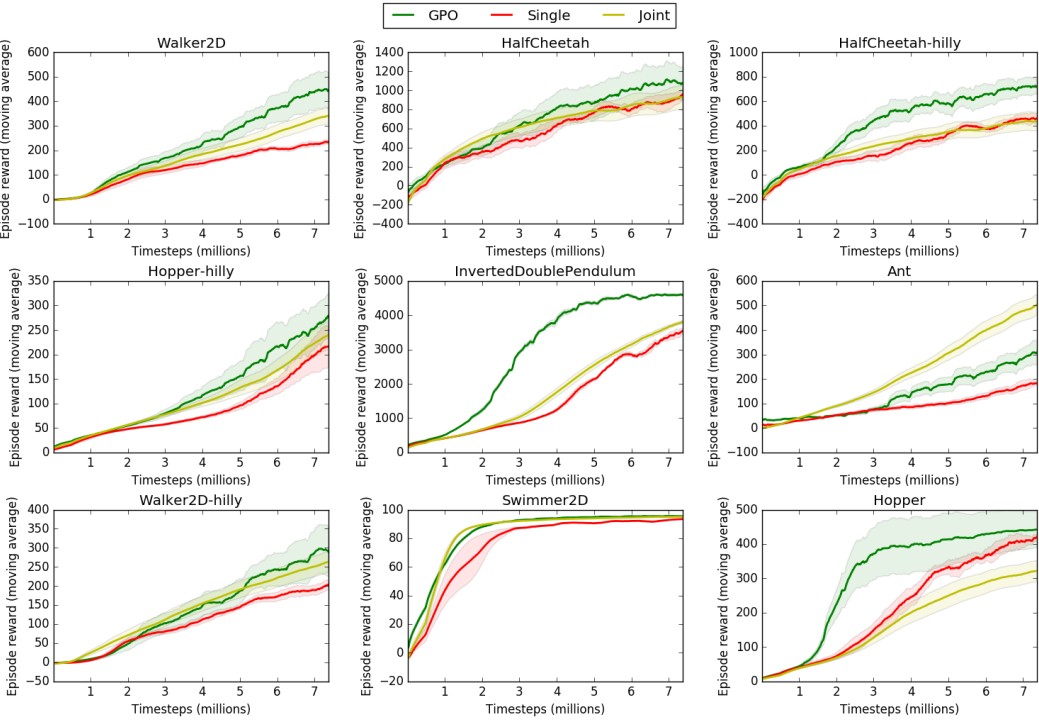

Figure 8: Performance of GPO and baselines on MuJoCo environments using A2C.

|  | GPO | Single | Joint |
|---|---|---|---|
| Walker2d | **444.7 ± 69.39** | 233.98 ± 7.9 | 340.75 ± 33.59 |
| HalfCheetah | **1071.49 ± 179.98** | 956.84 ± 54.12 | 930.17 ± 123.83 |
| HalfCheetah-hilly | **719.39 ± 63.74** | 460.59 ± 43.2 | 434.48 ± 75.24 |
| Hopper-hilly | **279.11 ± 40.28** | 216.43 ± 39.78 | 240.26 ± 28.67 |
| InvertedDoublePendulum | **4589.9 ± 45.37** | 3545.43 ± 83.63 | 3802.8 ± 50.56 |
| Ant | 308.67 ± 49.63 | 182.61 ± 10.39 | **503.64 ± 42.01** |
| Walker2d-hilly | **289.51 ± 56.9** | 203.74 ± 12.77 | 263.49 ± 27.33 |
| Swimmer | **95.43 ± 0.11** | 93.43 ± 0.1 | 95.05 ± 0.05 |
| Hopper | **441.79 ± 47.21** | 421.06 ± 9.1 | 321.48 ± 30.92 |

Table 2: Mean and standard-error for final performance of GPO and baselines using A2C.

## 6.2 RLLAB VS. GYM ENVIRONMENTS

We use the OpenAI rllab framework, including the MuJoCo environments provided therein, for all our experiments. There are subtle differences in the environments included in rllab and OpenAI Gym repositories [2] in terms of the coefficients used for different reward components and aliveness bonuses. In Figure 9, we compare GPO and `Joint` baseline using PPO on three Gym environments, for two different time-horizon values (512, 1024).

---

[2] `github.com/rll/rllab/blob/master/rllab/envs/mujoco/walker2d_env.py`
`github.com/openai/gym/blob/master/gym/envs/mujoco/walker2d.py`

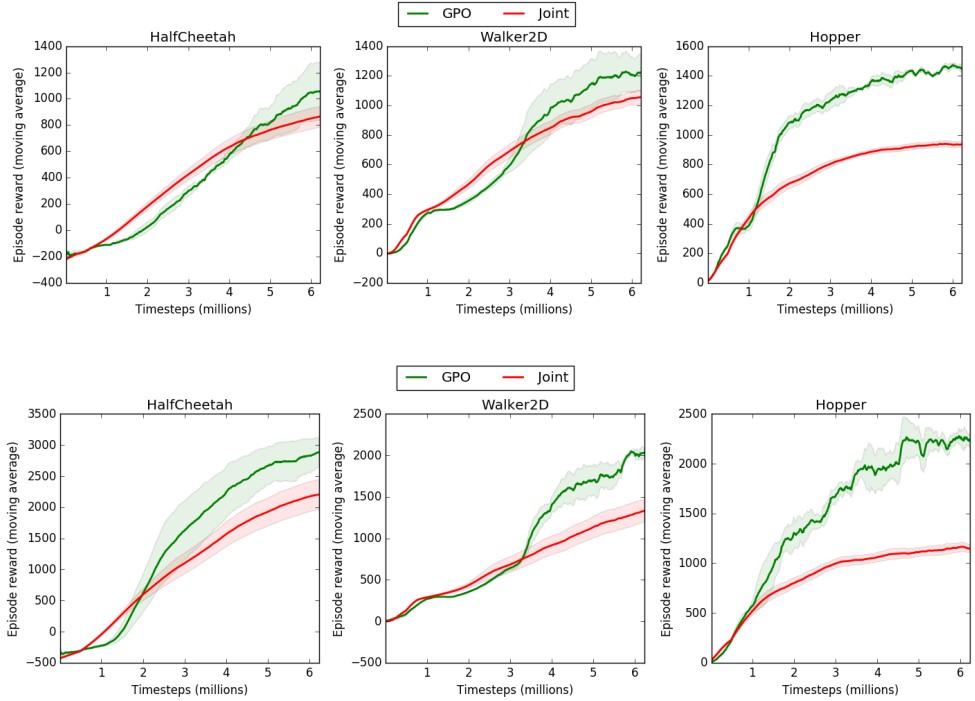

Figure 9: Performance of GPO and `Joint` baseline on MuJoCo environments from OpenAI Gym. Top row uses time-horizon of 512 steps; bottom row uses time-horizon of 1024 steps.

### 6.3 CROSSOVER IMPLEMENTATION DETAILS

The crossover stage is divided into two phases - a) training the binary policy ($\pi_S$) and 2) imitation learning. For training $\pi_S$, the dataset consists of trajectories from the parents involved in the crossover. We combine the trajectories from the parents' previous mutation phase, and filter based on trajectory rewards. For our experiments, we select top 60% trajectories from the pool, although we did not find the final GPO performance to be very sensitive to this hyperparameter. We do 100 epochs of supervised training with Adam (mini-batch=64, learning rate=5e-4) on the loss defined in Equation 1.

After training $\pi_S$, we obtain expert trajectories (5k transitions) from $\pi_H$. Imitation learning is done in a loop which is run for 10 iterations. In each iteration $i$, we form the dataset $\mathcal{D}_i$ using existing expert trajectories, plus new trajectories (500 transitions) sampled from the student (child) policy $\pi_c^{(i)}$ with the actions labelled by the expert. Therefore, the ratio of student to expert transitions in $\mathcal{D}_i$ is linearly increased from 0 to 1 over the 10 iterations. We then update the student by minimizing the KL divergence objective over $\mathcal{D}_i$. We do 10 epochs of training with Adam (mini-batch=64, learning rate=5e-4) in each iteration.

#### 6.3.1 OTHER HYPERPARAMETERS

- Horizon (T) = 512

- Discount ($\gamma$) = 0.99

- PPO epochs = 10

- PPO/A2C batch-size (GPO, Single) = 2048

- PPO/A2C batch-size (Joint) = 16384

## 6.4 MISCELLANEOUS

### 6.4.1 WALL-CLOCK TIME

Table 3 shows the wall-clock time (in minutes), averaged over 3 runs, for GPO and the `Joint` baseline. All runs use the same number of simulation timesteps, and are done on an Intel Xeon machine [3] with 12 cores. For GPO, Mutate takes the major chunk of total time. This is partially due to the fact that data-sharing between ensemble policies leads to communication overheads. Having said that, our current implementation based on Python Multiprocessing module and file-based sharing in Unix leaves much on the table in terms of improving the efficiency for Mutate, for example by using MPI. `Joint` trains 1 policy with $8\times$ the number of samples as each policy in the GPO ensemble. However, sample-collection exploits 8-way multi-core parallelism by simulating multiple independent environments in separate processes.

| | GPO | | | | Joint |
|---|---|---|---|---|---|
| | Total | Mutate | Crossover | Select | |
| Walker2D | 79.53 | 51.60 | 12.93 | 15.0 | 47.17 |
| Half-Cheetah | 78.80 | 50.0 | 12.73 | 16.07 | 46.16 |
| Half-Cheetah-hilly | 79.80 | 49.73 | 12.60 | 17.47 | 46.33 |
| Hopper-hilly | 85.60 | 53.27 | 12.93 | 19.40 | 42.83 |
| InvertedDoublePendulum | 71.20 | 44.53 | 13.87 | 12.80 | 35.67 |
| Ant | 101.80 | 61.0 | 19.60 | 21.20 | 64.33 |
| Walker2d-hilly | 84.0 | 52.80 | 14.67 | 16.53 | 44.58 |
| Swimmer | 79.07 | 49.67 | 12.53 | 16.87 | 50.0 |
| Hopper | 82.27 | 51.67 | 15.0 | 15.60 | 41.92 |

Table 3: Average wall-clock time (in minutes) for GPO and Joint.

### 6.4.2 SCALABILITY

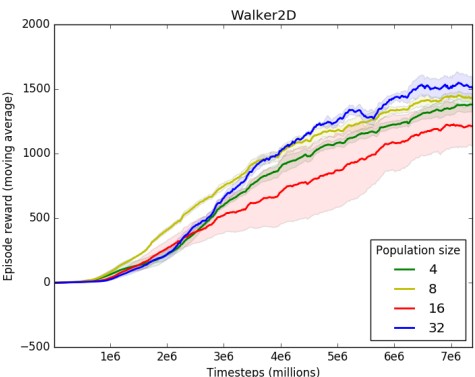

Figure 10: Effect of GPO population size when using same number of timesteps.

In Figure 7, we ran Walker2D with different population-size values, and compared performance. All policies used the same batch-size and number of gradient steps, making the total number of simulation timesteps grow with the population-size. In Figure 10, we show results for the same environment but reduce the batch-size for each gradient step in proportion to the increase in population-size. Therefore, all experiments here use equal simulation timesteps. We observe that the sample-complexity for a population of 32 is quite competitive with our default GPO value of 8.

---

[3] Intel CPU E5-2620 v3 @ 2.40GHz

