# OpenReview forum: "Policy Optimization by Genetic Distillation "
_ICLR.cc/2018/Conference — Accept (Poster)_

### Official Review · AnonReviewer2 · 2017-11-23
**novel but incomplete**

**Rating:** 8
**Confidence:** 5

**Review:**

This is a highly interesting paper that proposes a set of methods that combine ideas from imitation learning, evolutionary computation and reinforcement learning in a novel way. It combines the following ingredients:
a) a population-based setup for RL
b) a pair-selection and crossover operator
c) a policy-gradient based “mutation” operator
d) filtering data by high-reward trajectories
e) two-stage policy distillation

In its current shape it has a couple of major flaws (but those can be fixed during the revision/rebuttal period):

(1) Related work. It is presented in a somewhat ahistoric fashion. In fact, ideas for evolutionary methods applied to RL tasks have been widely studied, and there is an entire research field called “neuroevolution” that specifically looks into which mutation and crossover operators work well for neural networks. I’m listing a small selection of relevant papers below, but I’d encourage the authors to read a bit more broadly, and relate their work to the myriad of related older methods. Ideally, a more reasonable form of parameter-crossover (see references) could be compared to -- the naive one is too much of a straw man in my opinion. To clarify: I think the proposed method is genuinely novel, but a bit of context would help the reader understand which aspects are and which aspects aren’t.

(2) Ablations. The proposed method has multiple ingredients, and some of these could be beneficial in isolation: for example a population of size 1 with an interleaved distillation phase where only the high-reward trajectories are preserved could be a good algorithm on its own. Or conversely, GPO without high-reward filtering during crossover. Or a simpler genetic algorithm that just preserves the kills off the worst members of the population, and replaces them by (mutated) clones of better ones, etc.

(3) Reproducibility. There are a lot of details missing; the setup is quite complex, but only partially described. Examples of missing details are: how are the high-reward trajectories filtered? What is the total computation time of the different variants and baselines? The x-axis on plots, does it include the data required for crossover/Dagger? What are do the shaded regions on plots indicate? The loss on \pi_S should be made explicit. An open-source release would be ideal.

Minor points:
- naively, the selection algorithm might not scale well with the population size (exhaustively comparing all pairs), maybe discuss that?
- the filtering of high-reward trajectories is what estimation of distribution algorithms [2] do as well, and they have a known failure mode of premature convergence because diversity/variance shrinks too fast. Did you investigate this?
- for Figure 2a it would be clearer to normalize such that 1 is the best and 0 is the random policy, instead of 0 being score 0.
- the language at the end of section 3 is very vague and noncommittal -- maybe just state what you did, and separately give future work suggestions?
- there are multiple distinct metrics that could be used on the x-axis of plots, namely: wallclock time, sample complexity, number of updates. I suspect that the results will look different when plotted in different ways, and would enjoy some extra plots in the appendix. For example the ordering in Figure 6 would be inverted if plotting as a function of sample complexity?
- the A2C results are much worse, presumably because batchsizes are different? So I’m not sure how to interpret them: should they have been run for longer? Maybe they could be relegated to the appendix?

References:
[1] Gomez, F. J., & Miikkulainen, R. (1999). Solving non-Markovian control tasks with neuroevolution.
[2] Larranaga, P. (2002). A review on estimation of distribution algorithms.
[3] Stanley, K. O., & Miikkulainen, R. (2002). Evolving neural networks through augmenting topologies.
[4] Igel, C. (2003). Neuroevolution for reinforcement learning using evolution strategies.
[5] Hausknecht, M., Lehman, J., Miikkulainen, R., & Stone, P. (2014). A neuroevolution approach to general atari game playing.
[6] Gomez, F., Schmidhuber, J., & Miikkulainen, R. (2006). Efficient nonlinear control through neuroevolution.


Pros:
- results
- novelty of idea
- crossover visualization, analysis
- scalability

Cons:
- missing background
- missing ablations
- missing details

[after rebuttal: revised the score from 7 to 8]

---

> ### Author Response · Authors · 2017-12-28
> **Response to AnonReviewer2. Thank you for your comments!**
>
> 1- Concerning “Missing background” : Thank you for pointing us to relevant literature on neuroevolution algorithms for reinforcement learning. Previously, we had only covered NEAT and evolutionary strategies (ES, CMA-ES), but we have expanded our background section to include HyperNEAT, CoSyNE, SANE and ESP, since all these neuroevolution algorithms have been successfully applied to RL problems. Please see section 2.3. We have also include a recent work by Lehman et. al on “safe mutation” for genetic algorithms.
>
> 2- Concerning “Missing ablations” : We have added a section (4.4) on ablation studies for GPO. We consider the following 3 crucial components of GPO - 1) State-space crossover 2) Selection of candidates with highest fitness 3) Data-sharing for policy gradient RL during the mutation phase. To understand the behavior of these components, we compare performance when each of them is used in isolation for policy optimization. We further experiment with the scenario when one component is removed from GPO and the other two are used. This gives us a total of 6 algorithms which we plot along with GPO and our Single baseline in Figure 5 in the revision. It helps to define what we mean by a component “not being used”. For crossover, it means that we create the offspring by using the network parameters of the stronger parent; for selection, it means that we disregard the fitness of candidates and select the population for the next generation at random; for data-sharing, it means that policies in the ensemble don’t share samples from other similar policies for PPO (or A2C) during mutation.
>
> 3- Concerning “Reproducibility” : We have added a section (6.3) on implementation details for the crossover step. To train the binary policy (Equation 1 in the revision), we reuse the trajectories from the parents’ previous mutation phase rather than generating new samples. We filter the trajectories based on trajectory reward (sum of the individual reward at each transition in the trajectory). For our experiments, we simply prune the worst 40% trajectories from the dataset. We did not find the final GPO performance to be very sensitive to the % threshold hyperparameter. We will release our code on Github very soon.
>
> 4- Concerning other missing details : For computation time, we include a section (6.4.1) on wall-clock time. For all the environments, we compute the time for GPO and break it down into crossover, selection and mutation phases. We compare the time with our strongest baseline - “Joint”. The x-axis on all the plots of episode-reward vs. timesteps includes the data required for crossover/Dagger, and the shaded region indicates the standard-error of the performance with different random seeds. These details, along with the loss on \pi_S (Equation 1), are in the revision.
>
> 5- Concerning “Minor points”:
>
> [Scalable selection] Yes, for a population of n, comparing all nC2 pairs is prohibitively expensive when n is large. This was indeed the case when we ran with a population size of 32 for the scalability plot (Figure 7). Our solution was to prune the population to k by probabilistic sampling (probability = fitness), and then run selection over kC2. Looking for more sophisticated and scalable alternatives is interesting future work.
>
> [Lack of diversity] Yes, we did observe that maintaining a diverse population was challenging after 3-4 rounds of GPO (algorithm 1). We did some preliminary investigation with the Hopper environment, where we believed that some policies in the GPO ensemble were getting stuck in local minima, making the overall learning slow. We increased the randomness in the selection phase and found learning to proceed at a much more rapid pace. We need to explore this further.
>
> [Language at end of section 3] We have modified the section to include details on the fitness used by our experiments. Rather than dynamically adapting the weight of performance vs. KL fitness over the rounds of GPO, our current implementation puts all the weight on performance for all rounds, and relies on the randomness in the starting seed for different policies in the ensemble for diversity in the initial rounds.
>
> [Figure 6 with timesteps on x-axis] We have included this figure in Appendix 6.4.2. For the Walker environment, we observe that the sample-complexity for a population of 32 is quite competitive with our default GPO value of 8.
>
> [A2C results] A2C runs use the same batchsize as the PPO. We believe that the KL penalty in PPO prevents (possibly destructive) large updates to the policy distribution, and also the 10x more gradient steps in PPO allow for faster learning compared to A2C. A2C performance seems to be still improving when we end training for our experiments, and running them longer could see them match the PPO numbers. A2C results are moved to the Appendix in the revision.

---

> > ### Comment · AnonReviewer2 · 2018-01-02
> > **no more concerns**
> >
> > Thank you for thoroughly revising the paper, in particular, the new ablation studies are very insightful and substantially improved the paper -- and I'm surprised by how much of the contribution comes from data-sharing as compared to the other ingredients.
> >
> > (I raised my review score accordingly)

---

### Official Review · AnonReviewer1 · 2017-11-28
**Could equally well be titled "crossover by distillation"**

**Rating:** 6
**Confidence:** 4

**Review:**

The authors present an algorithm for training ensembles of policy networks that regularly mixes different policies in the ensemble together by distilling a mixture of two policies into a single policy network, adding it to the ensemble and selecting the strongest networks to remain (under certain definitions of a "strong" network). The experiments compare favorably against PPO and A2C baselines on a variety of MuJoCo tasks, although I would appreciate a wall-time comparison as well, as training the "crossover" network is presumably time-consuming.

It seems that for much of the paper, the authors could dispense with the genetic terminology altogether - and I mean that as a compliment. There are few if any valuable ideas in the field of evolutionary computing and I am glad to see the authors use sensible gradient-based learning for GPO, even if it makes it depart from what many in the field would consider "evolutionary" computing. Another point on terminology that is important to emphasize - the method for training the crossover network by direct supervised learning from expert trajectories is technically not imitation learning but behavioral cloning. I would perhaps even call this a distillation network rather than a crossover network. In many robotics tasks behavioral cloning is known for overfitting to expert trajectories, but that may not be a problem in this setting as "expert" trajectories can be generated in unlimited quantities.

---

> ### Author Response · Authors · 2017-12-28
> **Response to AnonReviewer1. Thank you for your comments!**
>
> 1- Concerning wall-time comparison: We have added a section in the Appendix (6.4.1) comparing wall-clock time for GPO and Joint. Both the algorithms are designed to use the multi-core parallelism on offer. We observe that GPO can be 1.5 to 2 times slower than Joint depending on the environment. Note that the timing numbers also depend on the number of iterations we run mutation (policy gradient) for before crossing over the policies, and we show the numbers for the default setting of these hyperparameters for all our experiments. For GPO, Mutate takes a good portion of the overall time due to communication overheads caused by data-sharing between policies. The crossover step takes moderate amount of time. We believe this is due to the following reasons - 1) for learning the binary policy (Equation 1), we reuse the trajectories from the parents’ previous mutation phase rather than generating new samples; 2) the losses in Equation 1. and 2. are not minimized to convergence since the optimization (first-order, Adam) is only run for certain number of epochs. We provide the exact details in a new section in the Appendix (6.3); (3) the crossover phase is parallelized, i.e. once the parents are decided by the selection step, each crossover is done in an independent parallel process.
>
> 2- Concerning use of term behavioral cloning: We completely agree with the reviewer that it’s imperative that we use crisp terminology. To that end, in section 3.2, where we first mention using imitation learning for the crossover, we expand on the differences between flavors of imitation learning (a.k.a behavioral cloning and inverse RL), and explicitly say that all our references to imitation learning signify behavioral cloning.
>
> 3- Concerning using “crossover by distillation”: We agree with the reviewer in that the high-level objective of the crossover step is to “distill” the knowledge from the parent policy networks into the offspring network. However, we believe that there are two main differences between the distillation network proposed in [1] and our procedure for crossover. Firstly, in [1] the soft targets for training the offspring network are computed using the arithmetic (or geometric) mean of the temperature-controlled outputs from parent networks. The argument is that different parent networks trained on similar data for similar amount of time represent different local minima point on the loss surface, and averaging leads to better generalization. In contrast, the parent policies in GPO have (possibly) visited disparate regions of the state-space and have (possibly) been trained on dissimilar data. Therefore, rather than averaging the output of the parents, we train another policy \pi_S to output the weighting, and do a weighted average. Secondly, the distillation network in [1] was trained for speech and object recognition tasks which do not have a temporal nature. However, the supervised training of the offspring in GPO should account for the compounding errors in the performance of the trained policy in areas of state-space different from the training data. Therefore, we add DAgger training to our crossover step, making it further different from vanilla distillation.
>
> [1] Hinton et al., Distilling the Knowledge in a Neural Network

---

### Official Review · AnonReviewer3 · 2017-11-29
**The title and motivation is somehow missing leading. The proposed method has no sound explanation. The experiment does not support the method well.**

**Rating:** 3
**Confidence:** 4

**Review:**

This paper proposes a genetic algorithm inspired policy optimization method, which mimics the mutation and the crossover operators over policy networks.

The title and the motivation about the genetic algorithm are missing leading and improper. The genetic algorithm is a black-box optimization method, however, the proposed method has nothing to do with black-box optimization.

The mutation is a method to sample individual independence of the objective function, which is very different with the gradient step. Mimicking the mutation by a gradient step is very unreasonable.

The crossover operator is the policy mixing method employed in game context (e.g., Deep Reinforcement Learning from Self-Play in Imperfect-Information Games, https://arxiv.org/abs/1603.01121 ). It is straightforward if two policies are to be mixed. Although the mixing method is more reasonable than the genetic crossover operator, it is strange to compare with that operator in a method far away from the genetic algorithm.

It is highly suggested that the method is called as population-based method as a set of networks is maintained, instead of as "genetic" method.

Another drawback, perhaps resulted from the "genetic algorithm" motivation is that the proposed method has not been well explained. The only explanation is that this method mimics the genetic algorithm. However, this explanation reveals nothing about why the method could work well -- a random exploration could also waste a lot of samples with a very high probability.

The baseline methods result in rewards much lower than those in previous experimental papers. It is problemistic that if the baselines have bad parameters.
1. Benchmarking Deep Reinforcement Learning for Continuous Control
2. Deep Reinforcement Learning that Matters

---

> ### Author Response · Authors · 2017-12-28
> **Response to AnonReviewer3. Thank you for your comments!**
>
> 1- Concerning title and missing motivation: The reviewer is correct in pointing out that genetic algorithms (GA) fall into the category of black-box optimization techniques. Their lack of exploiting the structure in the underlying tasks, e.g. the temporal nature in RL, explains their limited success in deep learning. Black-box techniques have been able to solve some RL problems, for example in [1] and most recently in [2], but with unsatisfactory sample-complexity. Our goal with GPO was to buy the philosophy of genetic operators - mutation, selection and crossover - from GA, and marry it with model-free policy-gradient RL algorithm to achieve good sample complexity. We believe that the connection to GA is helpful because it may be possible to apply the myriad of advanced enhancements for general GA (Section 3.1 in [2]) to our policy optimization algorithm as well. For example, techniques to obtain quality diversity in GA population could be helpful for efficient exploration in large state-spaces. At the same time, using policy gradients as a plug-and-play component in our genetically-inspired algorithm enables us to exploit advances in policy gradients; see, for instance, the difference in GPO performance with PPO compared to A2C.
>
> There is prior work on opening up the GA/ES “black-box” to obtain improved performance and stability for RL. For example, in [3], the authors suggest replacing the random mutations with perturbations guided by the gradients of the neural network output. A related idea was presented in [6]. [7] modifies the fitness function used in selection to aid exploration. We have updated section 2.3 with more neuroevolution algorithms which have been adjusted to work in the RL setting.
>
> 2- Concerning lack of explanation for why the method works: The fact that our algorithm is not “black-box” enables us to investigate the sources of improvement. Firstly, as we show in Section 4.2 through experimentation, the crossover operator is able to transfer positive behavior from the parent policies to the offspring policy. Secondly, we do mutation through tried-and-tested algorithms like PPO/A2C and take the empirical success that they have enjoyed. Thirdly, our selection operator maintains high performance policies in the population. We believe the overall GPO performance is a culmination of these components. We have added a section (4.4) on ablation studies for GPO.
>
> 3- Concerning baseline performance not same as other papers: We use the MuJoCo environments included as part of rllab [4]. The environments provided in the open-source release vary in their parameters from what the authors used for the paper (https://github.com/rll/rllab/issues/157), and therefore it’s hard to replicate their exact numbers. Regarding the numbers in [5], please note that their evaluation is done with the Gym MuJoCo environments, which again differ from rllab MuJoCo in terms of parameters like coefficients for rewards, aliveness bonus etc. For completeness, we ran GPO on Gym MuJoCo environments and compared to Joint. We have added Appendix 6.2 for this. We also had a discussion with the authors of [5] on the variance between baselines numbers for different codebases (rllab, openAIbaselines etc.). See Figure 6. in [5] for reference where “Duan 2016” is the rllab framework we use. We believe that factors such as value function approximation (Adam vs. LBFGS), observation/reward normalization method etc. lead to appreciable variation in baseline performance across codebases. Importantly, all these factors remain constant between GPO and the baselines for our results. Our baselines are very close to the rllab baselines (Figure 29.) in [5].
>
>
> [1] Evolution Strategies as a Scalable Alternative to Reinforcement Learning
> [2] Deep Neuroevolution: Genetic Algorithms Are a Competitive Alternative For Training Deep Neural Networks for Reinforcement Learning
> [3] Safe Mutations for Deep and Recurrent Neural Networks through Output Gradients
> [4] Benchmarking Deep Reinforcement Learning for Continuous Control
> [5] Deep Reinforcement Learning that Matters
> [6] Parameter Space Noise for Exploration
> [7] Improving Exploration in Evolution Strategies for Deep Reinforcement Learning via a Population of Novelty-Seeking Agents

---

> > ### Comment · AnonReviewer3 · 2018-01-05
> > **Unsatisfied revision**
> >
> > I have read the rebuttal and the revised paper. However, some problems remain.
> >
> > The major problem is that the revised paper is still motivated from evolutionary algorithm, instead of the rationality of the algorithm itself. Questions like "why crossover can work" and "why population is good" are left unanswered. Actually, these questions to the evoultionary algorithms are also unanswered. It is very sad about the loss of rationality.
> >
> > Figure 7 is misleading. The x-axis should be the number of samples, instead of iterations. We can compare the iteration at 200 of population 16 with iteration at 100 of population 32. My observation is that population 16 is more sample efficient than population 32, and also 8 is better than 16, 4 is better than 8. Population is not effective, but the number of iterations has the key impact. Figure 10 also shows that the population size is not a positive factor, meanwhile, the modification of the batch-size introduces other effect in another dimension.
> >
> > Why the batch-size of Joint is 8 times more than GPO and Single? Does that means the number of gradient updates of Joint is the same as Single? If so, the Joint has a lower performance than it could be by using 8 times more updates.

---

> > > ### Author Response · Authors · 2018-01-05
> > > **Response to AnonReviewer3. Thank you for further comments!**
> > >
> > > 1- Concerning unanswered questions like “why crossover can work” : A large portion of the paper is devoted to motivating why state-space crossover --- using a binary selection policy and imitation learning --- is a more thorough approach to mixing parents policies compared to parameter-space crossover. Section 3.2 has the intuition; section 3.3.1 has the procedure and algorithm; section 4.2 has the experimental validation of the claims with performance numbers and t-SNE plots.
> > >
> > > 2- Concerning “why population is good” and Figure 7 interpretation : With Figure 7 (and/or Figure 10), we do not claim that increasing population size improves sample-complexity. Instead, Figure 7 shows that adding policies to the ensemble (and consequently using more simulation timesteps) improves performance. We believe that the state-space for the MuJoCo environments doesn’t have sufficient richness and diversity to study the potential benefit on sample-complexity with a large population. In our experiments, although we start different policies in the population with different random seeds, it’s highly likely that they explore overlapping regions. Therefore, we neither expect nor mention consistent improvement in sample-complexity with population-size with MuJoCo. The situation might be different in harder RL tasks (e.g. robotics manipulation, grasping) with more state-variability, where population constituents can explore disparate slices of the state-space. Notwithstanding task diversity, we show that having multiple policies in the population and letting them interact through the operators in GPO is beneficial -- our “Joint“ baseline represents the algorithm with population of 1, and in Figure 4 we compare GPO (population size = 8) favorably to it for most of the environments.
> > >
> > > 3- Concerning batch-size and gradient steps : The batch-size of Joint is 8x more than GPO/Single because we desire a baseline algorithm that trains a single policy, using the same number of timesteps as GPO/Single (which train 8 policies). As mentioned in Section 4.3, the number of gradient update steps is the same for Joint and Single, but each step in Joint uses 8x data, leading to improved performance. Empirically, we found larger batch-size to be better than using more gradient steps with smaller batch-size.

---

### Author Response · Authors · 2017-12-27
**General response to the reviewers**

We would like to thank the anonymous reviewers for their comments and constructive feedback. We address each reviewer's comments individually and summarize the major changes made in the revision here:

1. Expanded section 2.3 to include missing background and extra citations on application of neuroevolution to reinforcement learning.
2. Added more details to Section 3.3.1 on crossover between policies, along with a schematic diagram for better elucidation.
3. Added ablation studies (Section 4.4).
4. Added implementation details for reproducibility (Appendix 6.3).
5. Added wall-clock time comparison (Appendix 6.4.1).
6. Added experiments with environments from OpenAI Gym in addition to rllab (Appendix 6.2) for comparison. Our baseline results are comparable to those in previous papers using rllab.

All additions are highlighted using red-colored text in the revision.

---

### Decision · Program_Chairs · 2018-01-29
**ICLR 2018 Conference Acceptance Decision**

**Decision:**

Accept (Poster)

**Comment:**

At least two of the reviewers found the proposed approach novel and interesting and worthy of publication at ICLR. The reviewers raised concerns regarding the paper's terminology, which may lead to some misunderstanding. I agree that upon a quick skim, a reader may think that the paper performs the crossover operation outlined at the bottom right of Figure 1. Please consider improving the figure and the caption to prevent such a misunderstanding. You can even slightly change the title to reflect the policy distillation operation rather than naive crossover. Finally, including some more complex baselines benefits the paper. I am curious whether performing policy gradient on an ensemble of 8 policies + periodic removal of the bottom half of the policies will provide similar gains.